# VIVA Stent Preclinical Evaluation in Swine: A Novel Cerebral Venous Stent with a Unique Delivery System

**DOI:** 10.3390/jcm14134721

**Published:** 2025-07-03

**Authors:** Yuval Ramot, Michal Steiner, Udi Vazana, Abraham Nyska, Anat Horev

**Affiliations:** 1Department of Dermatology, Hadassah Medical Center, Jerusalem 9112001, Israel; 2Faculty of Medicine, Hebrew University of Jerusalem, Jerusalem 7661041, Israel; 3Independent Preclinical Consultant, Rehovot 7630511, Israel; 4LAHAV CRO, Kibbutz Lahav, D.N. Negev 85335, Israel; udiv@lahavcro.com; 5Independent Consultant in Toxicologic Pathology, Tel-Aviv University, Tel-Aviv 6997801, Israel; 6V-Flow 21 Ltd., Modi’in-Maccabim-Re’ut 7178594, Israel; 7Neurology Department, Soroka University Medical Center, Beer-Sheva 8410101, Israel

**Keywords:** venous stenting, VIVA Stent System, transverse sinus stenosis, venous sinus stenting, preclinical study, thrombogenicity

## Abstract

**Background:** Venous sinus stenting is a promising treatment for intracranial venous disorders, such as idiopathic intracranial hypertension and pulsatile tinnitus, associated with transverse sinus stenosis. The VIVA Stent System (VSS) is a novel self-expanding braided venous stent designed to navigate tortuous cerebral venous anatomy. This preclinical study assessed the safety, thrombogenicity, and performance of the VSS in a swine model. **Methods:** Fifteen swine underwent bilateral internal mammary vein stenting with either the VSS (*n* = 9) or the PRECISE^®^ PRO RX stent (*n* = 6, reference). Fluoroscopy and thrombogenicity assessments were conducted on the day of stenting, clinical pathology analysis was carried out throughout the in-life phase, and CT Venography was performed before sacrifice. Animals were sacrificed at 30 ± 3 or 180 ± 11 days post-stenting for necropsy and histological evaluation. **Results:** Fluoroscopic angiography confirmed the successful VSS deployment with complete venous wall apposition and no vessel damage. The VSS achieved the highest scores on a four-point Likert scale for most performance parameters. No thrombus formation was observed on either delivery system. CT Venography confirmed vessel patency, no stent migration, and complete stent integrity. Histopathology showed a mild, expected foreign body reaction at 30 days, which resolved by 180 days, indicating normal healing progression. Both stents showed increased luminal diameter and decreased wall thickness at 180 days, suggesting vessel recovery. No adverse reactions were observed in non-target organs. **Conclusions:** The VSS exhibited favorable safety, procedural performance, and thromboresistance in a swine model, supporting its potential clinical use for treating transverse sinus stenosis and related conditions.

## 1. Introduction

The development of endovascular stents has revolutionized the management of vascular disease, offering a minimally invasive alternative to open surgical procedures. Among the various clinical applications, venous sinus stenting has gained increasing attention, particularly in the treatment of idiopathic intracranial hypertension (IIH) and pulsatile tinnitus (PT). In many cases, both conditions are associated with transverse sinus stenosis (TSS) [1,2,3]. IIH is characterized by persistently elevated intracranial pressure (ICP) without a clear etiology, often leading to debilitating headaches, visual disturbances and in severe cases, permanent loss of vision [4]. PT results from turbulent blood flow in the venous sinuses adjacent to the auditory structures, leading to a debilitating perception of a rhythmic sound [5]. Venous sinus stenting has demonstrated promising clinical outcomes in addressing these conditions, significantly reducing ICP in IIH patients and eliminating the audible symptoms of PT [6].

Current commercial stents used for venous sinus stenting are largely the off-label adaptations of arterial stents, which often lack the specific flexibility, radial force, length, and re-sheathing ability required for safe and effective deployment in the cerebral venous sinuses [1,7]. Commonly used stents for venous sinus stenting include Zilver (Cook Medical, Bloomington, IN, USA) [8,9], Precise (Cordis, Miami Lakes, FL, USA) [10,11,12,13], Protégé (Medtronic, Minneapolis, MN, USA) [14], and Wallstent (Boston Scientific, Marlborough, MA, USA) [8,15], which are peripheral arterial stents. To address the anatomical and procedural challenges of treating transverse sinus stenosis [16], including the limitations of currently available arterial stents—such as excessive radial force, poor flexibility, and lack of re-sheathability—the VIVA Stent System (VSS) was developed as a dedicated, braided venous stent designed specifically for cerebral venous anatomy. It includes innovations such as rear crimping for improved trackability, controlled radial force, and a dedicated release mechanism to enhance procedural safety.

This Good Laboratory Practice (GLP)-compliant study was aimed to evaluate the safety and performance of the VSS in a swine model, utilizing fluoroscopy, thrombogenicity, clinical pathology analysis, computed tomography, angiography, and histopathological examinations to assess its performance and potential adverse effects. The PRECISE^®^ PRO RX stent, one of the most commonly used off-label stents for venous sinus stenting, was selected as a reference control. The swine model was selected for this study as swine are widely recognized for their anatomical and physiological similarities to humans, particularly their vascular anatomy and functionality [17]. The internal mammary vein was chosen due to its relative accessibility, consistent diameter, and size similarity to the human transverse sinus. Additionally, its anatomical location above the heart offers a hemodynamic environment more comparable to that of the human cerebral venous system, thereby allowing for a more clinically relevant evaluation of the VIVA stent [18,19], and to the outer diameter of the VIVA and PRECISE^®^ PRO RX stents (i.e., 7 mm). While this model does not fully replicate the complex hemodynamic and anatomical characteristics of intracranial venous sinuses, it enables controlled and reproducible assessment of device safety, vascular response, and technical performance—key endpoints for preclinical evaluation under GLP conditions.

Despite the growing interest in venous sinus stenting, there is a lack of dedicated devices supported by rigorous preclinical evidence. This study was performed to address this gap by providing the first GLP-compliant evaluation of a cerebral venous stent specifically engineered for the anatomical and physiological constraints of the dural venous sinuses. Unlike prior studies that adapted arterial stents for off-label venous use, this work offers translational insight into the safety and functional performance of a purpose-built stent system.

## 2. Materials and Methods

### 2.1. Study Design

This study was designed in accordance with international regulatory guidelines for medical device evaluation and also adhered to the ISO 10993 standards—selection of tests for interactions with blood: part 4 for the thrombogenicity evaluation and tests for local effects after implantation [20] and part 6 for the assessment of the long-term local implant reactions [21].

This study was conducted at The Institute of Animal Research, Kibbutz Lahav, Israel. All animal handling and procedures were conducted in compliance with institutional and national animal welfare regulations (National Council for Animal Experimentation approval no. NPC-La-IL-2204-129-4, approved on 7 April 2022, and NPC-La-IL-2310-526-4, approved on 10 October 2023) and OECD GLP principles.

### 2.2. Test and Reference Items

The VSS is a braided, self-expanding venous stent made of nitinol, with a nominal diameter of 7.0 mm and a length of 50 mm. The stent is crimped at the rear section of the dedicated delivery catheter, enhancing trackability through tortuous venous anatomy. It can be advanced to the tip and deployed with re-sheathability of up to 90% of its length, allowing repositioning. The system includes a release button to prevent premature deployment. The VSS was designed with reduced radial force to minimize trauma to the venous wall while maintaining sufficient expansion to restore luminal patency. The stent and delivery system are compatible with a 7F introducer sheath.

The PRECISE^®^ PRO RX stent (Cordis Ltd., Miami Lakes, FL, USA), a commonly used carotid stent, was selected as the reference device. Although the PRECISE stent was originally designed for carotid arteries, it was selected as the comparator because it is among the most widely used devices in off-label clinical practice for cerebral venous sinus stenting. Thus, it served as a relevant real-world reference for procedural performance and biological response.

### 2.3. Animal Model and Grouping

Fifteen female domestic (a breed of Landrace and Large White) pigs (60–68 Kg in body weight) were used in this study. The animals were randomly divided into two groups (via https://www.randomlists.com):
Test group (*n* = 9): implanted with the VIVA stent.Reference group (*n* = 6): implanted with the PRECISE^®^ PRO RX stent.

Each animal underwent bilateral stent placement in the right and left internal mammary veins (Table 1).

The sample size (*n* = 15) was selected based on preclinical feasibility, ethical considerations, and GLP guidelines. This study was not designed for formal hypothesis testing, and no statistical power calculations were performed. The primary aim was to qualitatively and descriptively assess device safety, technical performance, and biological response. The group allocation (VIVA Stent System = 9; comparator stents = 6) reflects the need for broader characterization of the investigational device while still allowing for reference comparison to existing technologies. In designing the study, we adhered to the FDA’s 2023 guidance on General Considerations for Animal Studies Intended to Evaluate Medical Devices [22], which recommends using the minimum number of animals necessary to obtain predictive and meaningful results. Since there are currently no approved venous stents designed specifically for cerebral sinus applications, we selected an arterial stent that is commonly used off-label in this context as a comparator. Acknowledging that this is not an ideal match and prioritizing animal welfare, we limited the number of animals implanted with the comparator device accordingly.

### 2.4. Procedure

#### 2.4.1. Preoperative Care and Anesthesia

Anti-platelet therapy began 5–7 days prior to the procedure with Aspirin (100 mg) and Plavix (75 mg) PO once daily, simulating as closely as possible the conditions of clinical use. Animals fasted for 12 h pre-procedure, with water available ad libitum. Anesthesia was induced with intramuscular ketamine (10 mg/kg) and Xylazine (2 mg/kg), followed by isoflurane inhalation (1–5% in oxygen 100%) via face mask. Pre-medications included Cefazolin (intravenous, 2.5 g/animal), Marbocyl (intramuscular, 2 mg/kg), Synulox (intramuscular, 8.75 mg/kg), and Morphine (intramuscular, 30–35 mg/animal). Venous catheterization was performed, followed by Diazepam (10 mg/animal) intravenous induction and intubation with an appropriately sized endotracheal tube. General anesthesia was maintained with isoflurane (1–5% in oxygen 100%). The inguinal area was disinfected with 4% (*w*/*v*) SEPTAL SCRUB^®^ (Teva Pharmaceuticals Ltd., Tel-Aviv, Israel) and ethanol (70%) (MediMarket Ltd., Emek Hefer, Israel). Vital signs, including heart rate, SpO_2_, temperature, systemic arterial blood pressure, ECG, and end-tidal CO_2_, were monitored. Baseline blood samples were analyzed for CBC, biochemistry, coagulation profiles, and free hemoglobin levels.

#### 2.4.2. Surgical Access and Stent Deployment

A femoral venous approach was used under fluoroscopic guidance. The target internal mammary veins were accessed via a 7F introducer sheath and a 7Fr Judkins Right/Guiding catheter. A 0.014” guidewire was advanced to the deployment site. Heparin was administered intravenously to maintain an Activated Clotting Time (ACT) of 250–350 s. The VIVA Stent (test item) or PRECISE^®^ PRO RX stent (reference item) was deployed following ACT confirmation. Stent positioning was verified via fluoroscopy before the retrieval of the delivery system.

### 2.5. Thrombogenicity

The external blood-contacting surfaces of each delivery system were examined following retrieval and semi-quantitatively evaluated for thrombus formation using the ISO-10993-4-adopted [20] grading system (refer to Appendix A).

### 2.6. Usability

Three interventionalists assessed the VSS usability using a Likert-scale (1 = poor, 4 = very good). Parameters evaluated included system preparation, catheter compatibility between components, adjunct equipment compatibility, navigation, trackability, positioning, radiopacity, stent deployment, disengagement, re-sheathing, retraction, and overall ease of use. While inherently subjective, these ratings provide practical insights into device usability under conditions simulating real-life handling and were supplemented by objective endpoints including fluoroscopic tracking, implantation success, and procedural time.

### 2.7. Postoperative Monitoring

Aspirin (100 mg) and Plavix (75 mg) administration were continued throughout the study. Clexane (60 mg) was administered intramuscularly twice daily until Day 13. Animals were monitored for distress and neurological deficits and weighed weekly. Blood samples were collected for CBC, biochemistry, coagulation, and free Hb analyses on days 1, 14 ± 2, 30 ± 3, 60 ± 5, 90 ± 5, and 180 ± 11.

### 2.8. Terminal Procedures

On 30 ± 3 days and 180 ± 11 days, animals underwent CT-based angiographies of the internal mammary veins before sacrifice, necropsy, and the histopathology of stented vessel segments and non-target organs (the brain, lungs, heart, thymus, liver, kidneys, spleen, lymph nodes, and mammary glands).

#### 2.8.1. Histopathological Processing

Histopathological examination was performed at StageBio (Frederick, MD, USA). Stented mammary vein segments were embedded in spur resin. Three cross-sectional levels were captured per segment: center, distal (~0.5 cm from the stent’s edge), and proximal (~0.5 cm from the stent’s edge), with at least 1 cm between sections. Slides were stained with Hematoxylin and Eosin (H&E). Two additional reference sections were stained proximally and distally. Lung samples included two sections per lobe. The heart was sectioned at three levels: left ventricle/atria, right ventricle/atria, and septum. All slides were stained with H&E.

#### 2.8.2. Histopathological Evaluation

A board-certified veterinary pathologist (https://ebvs.eu/colleges/ECVP/members/prof-abraham-nyska, accessed on 23 May 2025) performed the histopathological evaluation. Parameters assessed included vascular injury, endothelial loss, stent strut position, neointimal coverage, luminal occlusion, inflammatory cell counts, and tissue response using a semi-quantitative ISO 10993-6 [21] grading approach (refer to Appendix A).

### 2.9. Statistical Analysis

Normality testing (Shapiro–Wilk) was conducted before statistical comparisons. If Gaussian distribution was confirmed (*p* > 0.01), variance-equality testing (F-test) preceded Student’s *t*-test for single comparisons. For multiple comparisons, Bartlett’s test confirmed equal variances before applying one-way ANOVA with Dunnett’s test.

If Gaussian distribution was not verified (*p* < 0.01), or when analyzing ordinal data, non-parametric tests (Mann–Whitney U test or Kruskal–Wallis test for multiple comparisons) were applied.

Minor language editing was performed and the Graphical Abstract was created using a generative AI tool (ChatGPT, OpenAI, GPT-4, March 2025 version), with final output reviewed and edited by the authors.

## 3. Results

### 3.1. Mortality and Early Elective Sacrifice

A single case of unexpected mortality occurred in the 30-day cohort. Animal DP-14515, deployed with the test stent, was found dead on Day 18. Gross pathology examination revealed pale complexion and diffuse pallor of the liver and spleen, alongside swelling and diffuse intramuscular hematomas at the right inguinal area in proximity to the venous access site. The cause of death as determined by the designated pathologist was procedure-related bleeding due to trauma caused by the percutaneous sheath insertion to the femoral vein. An additional animal (DP-14261) from the 30-day cohort, also implanted with the test stent, was euthanized on Day 11 due to its inability to bear weight on its hindlegs. At necropsy, swelling of the right inguinal area accompanied with hematoma within the muscle was noted, and was attributed, as mentioned above, to procedure-related muscular trauma due to the percutaneous sheath insertion as determined by the designated pathologist. The histopathological evaluation of both animals revealed no test stent-related abnormalities in the internal mammary veins or in non-target organs.

### 3.2. Clinical Observations

Partial food consumption and hind limb weight-bearing difficulties accompanied by occasional swelling and hematoma formation in the inguinal region were noted in most animals for 10 days post-procedure. These findings were associated with recovery from general anesthesia and femoral vein catheterization.

### 3.3. Body Weight Trends

A general trend of weight maintenance was observed during the first week post-procedure, followed by appropriate weight gain in all animals throughout the observation period.

### 3.4. Fluoroscopy-Based Morphometric Assessments

Fluoroscopic angiography of the left and right internal mammary veins confirmed successful stent deployment in all animals, with full stent expansion and complete venous wall apposition. No instances of vascular damage, dissection, perforation, or vasospasm were recorded.

A statistically significant increase in venous diameter following the deployment of the reference stent (*p* = 0.005, paired *t*-test) was evident. The diameter of the veins ranged prior to implantation from 3.80 to 6.81 mm, and following stent deployment from 4.82 to 7.00 mm. In contrast, the deployment of the test stent resulted in a statistically significant reduction in vein diameter (*p* = 0.003, paired *t*-test). The diameter of the veins ranged prior to implantation from 3.35 to 8.70 mm, and following stent deployment from 4.20 to 6.34 mm. When cases with venous diameters of above 7 mm (exceeding the maximal stent diameter) prior to stenting were excluded from the test stent group, a statistically significant increase in post-deployment diameter was observed (*p* = 0.02, paired *t*-test).

Stent shortening was observed in both groups but did not differ significantly between the test and reference stents. Previous studies have indicated that braided stents (e.g., the test stent) exhibit greater shortening than laser-cut stents (e.g., the reference stent) [23].

### 3.5. Computerized Tomography Evaluations

All stented vessels remained patent, with no evidence of migration, fracture, kinking, or compression. In four test stents (two animals, 30-day cohort), a potential space between the stent and vessel wall at the distal (caudal) end was noted, while all test stents presented complete wall apposition as evaluated by fluoroscopic angiography. These observations were microscopically correlated to regions lacking neointimal fibroproliferative reaction along the vascular wall circumference during the histopathological examination of the stented vessels’ transverse sections, with no evidence of thrombosis. This lack of neointimal coverage may be attributed to the physical structure of the stent strut (i.e., a braided stent) or to the natural anatomy of the blood vessels (i.e., the intersection regions of branched vessels) and may not reflect a pathologic condition. A study by Yamaguchi et al. (2006) investigated the vascular histological response to oversized self-expanding stents in the porcine venous system [24]. The study reported that, at one- and three-months post-implantation, histopathological evaluation revealed uniform neointimal hyperplasia over the stent struts. Interestingly, at the intersections of branched vessels with the stent-implanted segments, no neointimal proliferation was observed, and the branches remained patent even when crossed by multiple stent struts. These findings emphasize that certain vascular regions, particularly those involving natural bifurcations, may exhibit reduced neointimal coverage. Similarly, in a study conducted by Che et al. (2019), a novel braided venous stent demonstrated complete endothelialization without thrombosis after 90 days [25]. As with Yamaguchi’s findings, no neointimal coverage was observed at certain branch vessel sites, further highlighting the importance of carefully distinguishing between normal anatomical variations and true pathological findings.

### 3.6. Thrombogenicity Evaluation

The inspection of the retrieved test or reference delivery systems showed no thrombus formation on all external blood-contacting surfaces and was scored 0 according to the semi-quantitative scoring system recommended in ISO-10993-4 [20].

### 3.7. Usability and Functionality Evaluations

Across all categories, including navigation, trackability, compatibility to adjunct equipment (i.e., introducer sheath, guidewire), detachment of the stent from the stent holder, and its re-sheathing, the test stent received a rating of 4 (“very good and acceptable”), aside from three cases (in two animals), in which difficulties in detachment of the stent from the stent holder were noted, and this parameter was graded at 2 (“fair and acceptable”). This issue prompted a geometric modification of the grasper component of the delivery system, which was implemented in the subsequent stenting sessions.

### 3.8. Clinical Pathology Assessments

Clinical pathology analysis (i.e., hematology, biochemistry, and coagulation parameters as well as free hemoglobin (Free Hb)) did not indicate any clinically significant abnormalities.

### 3.9. Gross Pathology and Histopathological Findings

#### 3.9.1. Internal Mammary Vein Segments

On Day 30 ± 3, transverse sections from both test and reference groups exhibited minimal-to-mild (Grade 1) inflammatory response, indicative of foreign body reaction, in the intima and media, consisting of lymphocytes, macrophages, and giant cells (Figure 1A–D). The severity of inflammation was slightly greater in the test stent group, although this difference did not reach statistical significance. The examination of early elective sacrifice and unexpected mortality cases showed no evidence of severe inflammatory response, necrosis, or stent-related pathology, further supporting that these adverse outcomes were procedure-related rather than stent-induced.

Mild neointimal fibroproliferative response was observed in both groups of the 30-day cohort, covering all of the vessel wall circumference. In some test stent sections, neointimal coverage was slightly reduced (≤75% wall coverage, *p* = 0.01, Mann–Whitney test) particularly at the distal ends. These test stent sections with reduced neointimal coverage were noted in the animal that died prior to the scheduled termination time point and also corresponded to areas in which potential space between the stent struts and vessel wall had been noted in the pre-terminal CT scans of two test-item-treated animals.

Luminal occlusion was minimal in both groups, with ≤20% luminal narrowing observed, but was absent in cases of unexpected mortality and early elective sacrifice. No thrombosis, hemorrhage, or mineralization were evident.

On Day 180 ± 11, the foreign body response had completely subsided in all animals (Figure 1E–H). Mild intimal fibroproliferative reaction was evident in all sections of both study groups, with the complete coverage of the vessel wall generally seen, while sporadic cases of incomplete coverage were observed for both reference (≤75%) and test stents (50–75%), with no statistically significant differences observed between the groups.

In both groups, no luminal occlusion, chronic inflammation, excessive fibrosis, or pathological vascular remodeling was detected.

A comparison between the 30-day and 180-day cohorts demonstrated a progressive increase in luminal diameter (*p* < 0.05, Wilcoxon Rank-Sum test/*t*-test) and a decrease in vascular wall thickness in both groups, while only the former was statistically significant. These observations suggest that both test and reference stents facilitated adaptive vascular remodeling over time.

Comparing test and reference groups, the reference stent group had a larger luminal diameter on both Day 30 ± 3 and Day 180 ± 11 (*p* < 0.01, *t*-test/Wilcoxon Rank Sum test). This difference is likely attributable to higher radial force exerted by the reference stent [26], which is a laser-cut design with a greater expansion force, compared to the braided configuration of the test stent. However, this did not translate into increased vessel injury, thrombosis, or delayed healing in the test group.

#### 3.9.2. Non-Target Organs

Microscopic observation of non-target organs confirmed only incidental or procedure-related findings deemed unrelated to the test/reference item.

The most common gross finding noted during necropsy involved discolored areas in sampled mammary gland, observed in 5 of 15 animals across both groups; however, no microscopic pathological changes were evident, suggesting that this finding was insignificant.

A detailed microscopic examination identified bronchopneumonia in two reference stent animals, mild hepatic congestion in three test stent animals, and interstitial mononuclear cell infiltration in a single kidney sample from a test animal. These findings were all classified as incidental, with no evidence of systemic inflammatory or toxic effects related to the stents. A single reference stent animal exhibited increased cellularity in the mediastinal lymph nodes, but this was not associated with systemic inflammation or adverse tissue reactions. Minimal splenic congestion was observed in 14 of 15 animals, likely due to anesthesia-related circulatory shifts during pre-terminal imaging and sample collection.

Animal DP-14261, for whom early sacrifice was applied, showed the gross evidence of hematoma in the right inguinal muscle, which was microscopically correlated to chronic inflammation associated with necrosis and mineralization. These findings further demonstrated the impact of femoral vein catheterization, and were correlated to clinical observations of motor deficits.

## 4. Discussion

The present study evaluated the safety and performance of the VSS in a swine model, focusing on its usability, thrombogenicity, vascular response, and long-term biocompatibility. The study findings demonstrated that the VSS exhibits favorable safety and performance characteristics, with a well-tolerated deployment procedure, no thrombogenicity, and progressive and controlled neointimal response over time. Previous research showed that venous sinus stenting has been established as an effective treatment in patients with intracranial venous disorders, reinforcing the potential of VSS as a viable option for treating intracranial venous disorders [27,28].

Primary technical success was indicated by the successful deployment of the test stents in the internal mammary veins with full venous wall apposition, and no damage, dissection, perforation, or vasospasm to the veins was revealed by intra-procedural fluoroscopy-based evaluations. Pre-terminal computerized tomography-based angiographies of the test stents indicated complete vessel patency as well as complete stent integrity with no fracture, kinks, or compression and no stent migration in all study animals. The examination of the test item’s delivery system for thrombogenicity indicated no presence of thrombus on the delivery system’s external blood-contacting surfaces, thus rendering the VSS as non-thrombogenic under the conditions herein described.

The secondary technical success of the VSS was indicated by the usability and functionality score as determined by three designated interventionalists. Aside from two cases graded at 2 (fair and acceptable) in the 180-day cohort, regarding the full detachment of the test stent from the delivery system, all evaluated cases in both cohorts were graded at 4 (very good and acceptable) for all categories. In the course of the initial stenting procedures of the animals assigned to the 180-day cohort, there were sporadic cases in which the test stents did not fully detach from the delivery system. No vascular damage occurred. A geometric modification was made to the grasper component of the delivery system, and these difficulties did not recur in the subsequent stenting procedures performed in animals assigned to the 30-day cohort. This highlights the importance of iterative design improvements in optimizing endovascular devices.

The histopathological analysis in this study confirmed controlled neointimal fibroproliferative reaction by 180 days, indicating an appropriate healing response. The inflammatory reaction observed at 30 days, characterized by macrophages, lymphocytes, and occasional giant cells, had completely resolved by 180 days, consistent with a typical foreign body response [29,30]. Importantly, no cases of excessive fibrosis, excessive neointimal hyperplasia, or luminal obstruction were observed, suggesting that the VSS supports stable vascular remodeling without the risk of in-stent restenosis. The progressive increase in luminal diameter and reduction in vascular wall thickness further support this conclusion. Notably, incomplete neointimal coverage was observed for test stents, mainly up to 30 days post-stenting, which is consistent with previous observations in venous stenting studies and is attributed to normal anatomical variations (e.g., vascular bifurcations) or to the stent’s physical structure (i.e., braided stent) rather than to pathological abnormalities [24,25].

Although venous sinus stenting has gained clinical traction, most devices currently used are off-label arterial stents not designed for the unique anatomical and physiological features of the cerebral venous system. These arterial stents pose significant challenges: their high radial force and rigidity can overstretch the thin-walled venous sinuses, potentially compromising vessel integrity and contributing to stent-adjacent stenosis. Their stiffness also hinders navigation through tortuous venous anatomy, often requiring intermediate catheters and complex manipulations that increase procedural time and risk. In contrast, the VSS was specifically developed to address these challenges. Its braided nitinol structure offers a lower, more controlled radial force, minimizing trauma to delicate venous walls. Rear-mounted stent crimping improves trackability, while up to 90% re-sheathability allows repositioning even after partial deployment. The integrated release button adds a safety mechanism that reduces the risk of premature deployment—features not present in currently approved or commonly used devices. The ideal venous stent must balance radial force with flexibility. The cerebral venous sinus system is essentially a low-pressure system with a pressure reflection of central venous pressure, that is, 2–6 mm Hg. The high radial force together with the increased stiffness of the arterial stents can overstretch the cerebral venous sinuses [31,32]. Overstretching of the vein can create a collapse of the vein adjacent to the stent. A major reason for stenting failure is stenosis proximal or distal to the stent. Stent-adjacent stenosis rates were found to be 14% to 18%, and in ~10% of cases, the stenosis is symptomatic and/or requires retreatment/extension [33]. Furthermore, navigating with the current off-label available stents through the tortuous venous anatomy is challenging due to the stiffness of the delivery system’s tip. The difficulty in navigating across a stenotic region can lead to abortion of the procedure and prevent treatment of the stenosis. To overcome this difficulty, usually manipulation using an intermediate catheter by several techniques is required first. These techniques are time-consuming and may increase the risk of complications [33]. This unique rear crimping design significantly enhances deliverability through complex venous pathways and may eliminate the need for intermediate catheters, thereby reducing procedural complexity and potential complications. Together, these design elements represent a meaningful advancement over currently available options and address a critical unmet need in the treatment of intracranial venous pathologies. The usability assessment, performed by three neuroradiological interventionalists, indicated that the VSS was easy to navigate and deploy.

Given the growing interest in venous sinus stenting as a potential treatment for idiopathic intracranial hypertension (IIH) and pulsatile tinnitus (PT), the development of a dedicated stent for this procedure has become increasingly important. The VSS addresses a significant unmet clinical need by offering design features optimized for the unique pathological and anatomical characteristics of the cerebral venous system. The VSS along with at least two other investigational devices, the River stent (Serenity Medical, Austin, TX, USA) [34], the first stent specifically designed for intracranial venous sinuses, and the BosStent (Sonorous Neurovascular, Lake Forest, CA, USA) [7] with varying properties, hold the potential to provide a durable solution for many patients suffering from debilitating symptoms and poor quality of life. The findings from this GLP-compliant preclinical study confirm that the VSS is safe, well tolerated, and effective in maintaining venous patency over six months. The absence of thrombosis, vessel injury, or adverse remodeling suggests that the VSS may provide a viable solution for intracranial venous stenosis requiring endovascular intervention. The unique design of the delivery system allows for easy tracking, as well as the ability for re-sheathing, together with the release button that prevents early detachment of the stent, enhancing both usability and safety. The braided structure appears to offer distinct advantages in terms of vessel wall adherence while avoiding the use of excessive radial force, which can potentially lead to better clinical outcomes. However, further clinical studies are needed to validate these results in human subjects. Given the increasing interest in venous sinus stenting as a potential treatment for IIH and PT, the VSS could address an unmet clinical need by providing a dedicated venous stent with optimized pathological and anatomical related properties.

While outcomes such as patency and the absence of thrombosis are expected for medical devices which have undergone extensive research and development leading to a GLP-compliant study, the scientific value of this work lies in demonstrating that a purpose-designed braided stent can achieve these endpoints while also addressing procedural limitations seen with off-label arterial stents. The ability to re-sheath up to 90% of the stent, the rear-mounted crimping for enhanced navigability, and the integration of a release-lock mechanism are novel features not previously tested in an in vivo setting. Importantly, this study also captured a real-time usability issue—the incomplete detachment of the stent from the delivery system—which was corrected through a design modification. This iterative learning under GLP conditions reflects a translational advancement in venous stent development, directly applicable to future first-in-human trials.

This study has several limitations that should be acknowledged. The sample size was relatively small (*n* = 15), which is typical for regulatory-compliant preclinical studies but limits the statistical power and generalizability of the findings. However, as the primary objective was regulatory-compliant early evaluation of safety and usability evaluation under GLP standards, the sample size and distribution were considered sufficient for early-stage assessment. Another limitation of the current study is the use of a subjective Likert scale-based grading approach for evaluating performance parameters. Although this reflects real-world operator experience, it lacks the reproducibility of quantitative metrics. In addition, the evaluation period, although extending up to six months, may not fully capture long-term device-related complications such as late in-stent stenosis. Future clinical studies will be essential to confirm the safety, usability, and efficacy of the VIVA Stent System in human patients, particularly in the complex and delicate cerebral venous anatomy.

## 5. Conclusions

In this GLP-compliant preclinical study, the VSS showed favorable safety and procedural performance outcomes in terms of vascular patency and healing, thromboresistance, and procedural usability parameters, with no evidence of thrombosis, adverse tissue remodeling, or stent migration over a 6-month follow-up period. Histological analyses confirmed controlled neointimal responses and resolution of foreign body inflammation, supporting the device’s long-term safety. These findings support the study’s initial hypothesis and highlight the VSS as a promising dedicated venous stent system for the treatment of transverse sinus stenosis and related intracranial venous pathologies, with the potential to overcome the limitations of currently used arterial stents. Further clinical evaluation is warranted to confirm these results in human subjects.

## Figures and Tables

**Figure 1 jcm-14-04721-f001:**
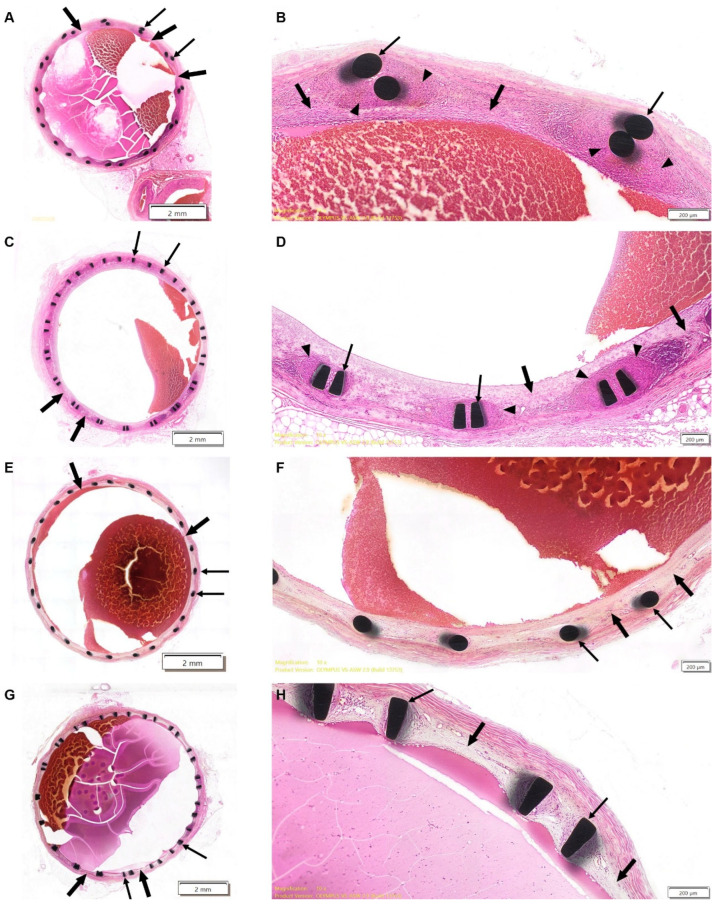
(**A**,**B**). Representative histological images of the proximal segment (~0.5 cm from the stent’s edge) of the VIVA Stent (test device) in Animal 14256, retrieved from the left internal mammary vein at 1 month post-implantation. (**A**) Low-magnification view demonstrating the overall stent structure and tissue response. (**B**) High-magnification view highlighting key histopathological features. Thin arrows indicate stent struts, while thick arrows mark the neointimal layer forming around the stent. The arrowhead denotes a foreign body reaction surrounding the struts. Red/purple luminal protrusions reflect post-mortem clot formation, which is a normal histological artifact. (**C**,**D**) Representative histological images of the proximal segment (~0.5 cm from the stent’s edge) of the PRECISE^®^ PRO RX Stent (reference device) in Animal 14255, retrieved from the left internal mammary vein at 1 month post-implantation. (**C**) Low-magnification view illustrating the overall stent structure and surrounding tissue response. (**D**) High-magnification view providing detailed visualization of the neointimal reaction. Thin arrows indicate stent struts, while thick arrows highlight the neointimal layer. The arrowhead denotes a foreign body reaction surrounding the struts. Red luminal protrusions correspond to post-mortem clot formation, a common histological artifact. (**E**,**F**) Representative histological images of the proximal segment (~0.5 cm from the stent’s edge) of the VIVA Stent (test device) in Animal 14114, retrieved from the right internal mammary vein at 6 months post-implantation. (**E**) Low-magnification view showing the overall stent structure and tissue integration. (**F**) High-magnification view detailing the neointimal response and luminal characteristics. Thin arrows indicate stent struts, while thick arrows highlight the neointimal layer. Red luminal protrusions represent post-mortem clot formation, a normal histological artifact. (**G**,**H**) Representative histological images of the proximal segment (~0.5 cm from the stent’s edge) of the PRECISE^®^ PRO RX Stent (reference device) in Animal 14121, retrieved from the left internal mammary vein at 6 months post-implantation. (**G**) Low-magnification view displaying the overall stent structure and surrounding tissue response. (**H**) High-magnification view providing detailed visualization of the neointimal reaction. Thin arrows indicate stent struts, while thick arrows highlight the neointimal layer. Red luminal protrusions correspond to post-mortem clot formation, a normal histological artifact.

**Table 1 jcm-14-04721-t001:** Study design.

Total Sample Size	Group No.	Per Group Sample Size	Animal ID	Test/Reference Item	Treatment	Scheduled Sacrifice
*n* = 15	1	*n* = 6	DP-14115	PRECISE^®^ PRO RX stent and delivery system (the reference item)	Deployment of either the VIVA or PRECISE^®^ PRO RX (per group assignment) into both the right and left internal mammary veins via transcatheter approach through the femoral vein	Day 183
			DP-14116			Day 191
			DP-14121			Day 185
			DP-14255 *			Day 31
			DP-14259			Day 33
			DP-14260			Day 33
	2	*n* = 9	DP-14114	VIVA Stent and delivery system (the test item)		Day 183
			DP-14117			Day 181
			DP-14120			Day 176
			DP-14196			Day 185
			DP-14256			Day 31
			DP-14258			Day 31
			DP-14261 **			Day 11
			DP-14514			Day 32
			DP-14515 ***			Day 18

* Deployment in the right mammary vein was not accomplished; ** Early elective sacrifice; *** Unexpected mortality.

## Data Availability

The original contributions presented in this study are included in the article/Appendix A. Further inquiries can be directed to the corresponding author(s).

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
