# Peer review of "VIVA Stent Preclinical Evaluation in Swine: A Novel Cerebral Venous Stent with a Unique Delivery System"

_jcm, 2025, doi:10.3390/jcm14134721_

Round 1
Reviewer 1 Report
Comments and Suggestions for Authors
Was there any evaluation of efficacy of P2Y12 inhibition prior to stent placement? This could affect the thrombogenicity assays.
Any further description/explanation for "a potential space between the stent and vessel wall and the distal (caudal) end"? Does this reflect poor wall apposition along the distal end of the stent? Did any stents require balloon angioplasty following deployment to improve wall apposition?
Please include some mention of the RIVER stent, the first stent to be made specifically for the purpose of intracranial venous sinus stenting. Has the VIVA stent been compared to the RIVER stent in benchtop testing?
Author Response
Was there any evaluation of efficacy of P2Y12 inhibition prior to stent placement? This could affect the thrombogenicity assays.
Reply: Thank you for bringing this up. Platelet inhibition test was not performed prior to stent placement as part of this study as it is not routinely used for cerebral venous stenting in humans, probably due to the low thrombotic events post-stenting related to the procedure (see Fargen. Venous stenting for idiopathic intracranial hypertension: lessons learned from a high-volume practice. J Neurointerv Surg. 2022 Jun;14(6):528-532 and Patsalides et al., The River study: the first prospective multicenter trial of a novel venous sinus stent for the treatment of idiopathic intracranial hypertension. J Neurointerv Surg. 2025 Apr 29:jnis-2024-022540. We will consider performing this assessment on our FIH study to provide excessive safety.
Any further description/explanation for "a potential space between the stent and vessel wall and the distal (caudal) end"? Does this reflect poor wall apposition along the distal end of the stent? Did any stents require balloon angioplasty following deployment to improve wall apposition?
Reply: Thank you for this comment. Two test item-deployed animals presented potential space between the stent and the vascular wall on both the left and right sides, at the distal (caudal) end. All deployed stents presented complete wall apposition (as evaluated based on fluoroscopic angiography, see section 3.4) with no application of balloon angioplasty. The addressed finding was evident upon pre-terminal computerized tomography (CT)-based evaluations, conducted at 31 days post-deployment.
Potential explanations for these observations can be the physical features of the braided stent or the natural anatomy of the blood vessels (i.e., intersection regions of branched vessels). These observations were microscopically correlated to regions lacking neointimal fibroproliferative reaction along the vascular wall circumference noted in the histopathological examination of the stented vessels, with no evidence of thrombosis. We have amended the relevant section in the manuscript accordingly.
Please include some mention of the RIVER stent, the first stent to be made specifically for the purpose of intracranial venous sinus stenting. Has the VIVA stent been compared to the RIVER stent in benchtop testing?
Reply: Thank you for this remark. We have included a mention of the River stent. Since the River stent is not an approved stent and is not commercially available, we did not perform a direct benchtop comparison between the VivaStent and the River stent, although such a comparison could indeed be valuable in the future.
Reviewer 2 Report
Comments and Suggestions for Authors
This manuscript titled "VIVA Stent Preclinical Evaluation in Swine: A Novel Cerebral Venous Stent with Unique Delivery System" has several unresolved issues and limitations that require significant revisions:
Venous sinus stenting is indeed a growing area of interest within neurointerventional treatment, especially in context of idiopathic intracranial hypertension and pulsatile tinnitus, this manuscript does not present genuinely original or clinically impactful advancement. Concept of stenting in cerebral venous disorders has already been explored extensively in both preclinical and clinical studies. Introduction of "VIVA Stent System"—although described as novel device—is not sufficiently differentiated from existing devices to warrant publication on basis of innovation alone. Authors fail to clearly articulate what unmet need VIVA Stent System addresses that currently approved stents do not.
In addition, manuscript does not define a clear knowledge gap or demonstrate how this study adds significant value to existing literature. Most of outcomes (e.g., vessel patency, no stent migration, mild inflammatory response) are expected baseline findings for any competent venous stent evaluated under GLP standards in animal models. Without comparative superiority data or unique mechanistic insights, study lacks scientific novelty.
The study design raises several methodological issues that limit its robustness:
Internal mammary vein of swine, while perhaps convenient, does not sufficiently replicate the complex hemodynamic and anatomical environment of human transverse sinus. This anatomical dissimilarity undermines translational value of findings.
Sample size is small (n=15), and allocation is uneven (VIVA Stent System = 9, comparator = 6), which introduces bias. No justification is provided for this distribution, and the manuscript does not report whether statistical power calculations were performed.
PRECISE® PRO RX stent, used as reference, is designed for carotid arteries—not cerebral venous sinuses—thus making it an inappropriate comparator. This weakens validity of performance and safety comparisons.
Use of subjective Likert scale for performance parameters is insufficient in a preclinical device evaluation. Absence of objective, quantitative metrics (e.g., radial force measurements, flow dynamics, endothelial coverage rate) limits reproducibility and weakens the scientific rigor.
Conclusion claims “excellent safety, performance, and thromboresistance” of VIVA Stent System, suggesting strong support for future clinical use. However, data presented are too limited to support such claims:
The term "excellent" is not substantiated by quantitative metrics. Conclusion extrapolates efficacy and safety from a small preclinical animal study to human application without appropriate caveats or discussion of limitations. Long-term biological compatibility, a crucial aspect of venous implants, is inadequately discussed given the limited observation time (180 days maximum).
This overreach in interpretation reflects a disconnect between results and conclusion, which may mislead readers about clinical readiness of device.
In this manuscript, there is no indication that authors have engaged with prior work on similar venous stents or cited key clinical trials in this space. High-quality device studies rely on detailed visual documentation, especially for angiographic and histopathologic validation.
Manuscript lacks novelty, uses suboptimal model and comparator, and overstates the significance of its findings. Methodological weaknesses, combined with exaggerated conclusion not supported by rigorous data. Authors should reconsider their experimental design, articulate clinical relevance more clearly, and adopt more cautious and evidence-driven approach to interpreting results.
Comments on the Quality of English LanguageThe English could be improved to more clearly express the research.
Author Response
This manuscript titled "VIVA Stent Preclinical Evaluation in Swine: A Novel Cerebral Venous Stent with Unique Delivery System" has several unresolved issues and limitations that require significant revisions:
Venous sinus stenting is indeed a growing area of interest within neurointerventional treatment, especially in context of idiopathic intracranial hypertension and pulsatile tinnitus, this manuscript does not present genuinely original or clinically impactful advancement. Concept of stenting in cerebral venous disorders has already been explored extensively in both preclinical and clinical studies. Introduction of "VIVA Stent System"—although described as novel device—is not sufficiently differentiated from existing devices to warrant publication on basis of innovation alone. Authors fail to clearly articulate what unmet need VIVA Stent System addresses that currently approved stents do not.
Reply: While we agree that venous sinus stenting is an increasingly established technique, we respectfully highlight that existing stents in clinical use are primarily arterial devices used off-label. These devices often exhibit excessive radial force, limited flexibility, and poor navigability through tortuous venous anatomy, which may contribute to procedural complications or suboptimal outcomes. It should be noted that there are still no approved stents designed specifically for cerebral sinus stenting.
In contrast, the VSS was explicitly engineered for the cerebral venous system. It is a dedicated braided venous stent featuring:
- Lower radial force to prevent overstretching of thin-walled venous structures;
- Rear crimping of the stent onto the delivery catheter for enhanced trackability through tortuous anatomy;
- Re-sheathability of up to 90% of its length for safer, more precise deployment;
- A delivery system with a release button, designed to prevent premature stent deployment.
These design attributes are not available in currently marketed stents, which were developed for arterial use. Furthermore, our preclinical data, including histopathology, usability scoring, and radiologic assessments, demonstrate favorable performance and safety. This supports the VSS as a clinically relevant and innovative step forward, particularly in addressing complications such as thrombosis, foreign body reaction, circumference covered by endothelium and stent-adjacent stenosis..
To clarify this distinction, we have revised the Introduction and Discussion sections to better articulate the clinical rationale and unmet need addressed by the VIVA Stent System.
In addition, manuscript does not define a clear knowledge gap or demonstrate how this study adds significant value to existing literature. Most of outcomes (e.g., vessel patency, no stent migration, mild inflammatory response) are expected baseline findings for any competent venous stent evaluated under GLP standards in animal models. Without comparative superiority data or unique mechanistic insights, study lacks scientific novelty.
Reply: We agree that baseline safety outcomes are expected for devices which had underwent extensive research and development leading to a GLP-compliant study; however, this study addresses a clear and specific knowledge gap: the lack of preclinical data on a cerebral venous stent system that was designed specifically for the unique biomechanical and anatomical requirements of the intracranial venous system. To our knowledge, this is the first GLP-compliant preclinical study to evaluate a dedicated braided venous stent with an integrated delivery system featuring rear crimping and re-sheathability up to 90%, tailored for the cerebral venous sinuses.
We have revised the Introduction and Discussion to clearly highlight this gap and to frame the study as a necessary translational step. We have also added clarifications in the Discussion emphasizing that the design-driven performance (trackability, deliverability, wall apposition, absence of trauma) and iterative engineering correction (grasper component fix) represent value-added contributions beyond routine safety confirmation.
The study design raises several methodological issues that limit its robustness:
Internal mammary vein of swine, while perhaps convenient, does not sufficiently replicate the complex hemodynamic and anatomical environment of human transverse sinus. This anatomical dissimilarity undermines translational value of findings.
Reply: We agree that no large animal model perfectly replicates the human transverse sinus in terms of anatomy and hemodynamics. The internal mammary vein in swine was selected as the implantation site primarily for its consistent caliber, accessibility, and relative size similarity to the human transverse sinus. Additionally, its anatomical location above the heart offers a hemodynamic environment more comparable to that of the human cerebral venous system, thereby allowing for a more clinically relevant evaluation of the VIVA stent. Importantly, it allowed for standardized assessment of device-related vascular injury, apposition, patency, and inflammatory response under GLP conditions, which were the main objectives of this study.
We have now clarified this rationale in the Methods and Discussion sections. We also acknowledge the limitation and have added a note that future clinical trials will be essential to confirm performance in the human intracranial venous system.
Sample size is small (n=15), and allocation is uneven (VIVA Stent System = 9, comparator = 6), which introduces bias. No justification is provided for this distribution, and the manuscript does not report whether statistical power calculations were performed.
Reply: We acknowledge that the overall sample size was modest (n=15) and that the allocation between the VIVA Stent System (n=9) and comparator stents (n=6) was not numerically balanced. However, in accordance with the FDA’s 2023 guidance on General Considerations for Animal Studies Intended to Evaluate Medical Devices, which emphasizes the use of the minimum number of animals necessary to yield predictive and interpretable outcomes, we designed the study to meet both scientific and ethical standards. Given that no approved venous stents exist for cerebral sinus applications, we selected an arterial stent commonly used off-label for venous sinus stenting as the comparator. Recognizing that this device is not an ideal match, and in consideration of animal welfare, we intentionally limited the number of animals receiving the comparator stent.
No formal statistical hypothesis testing or power calculations were conducted, as the sample size was determined in accordance with regulatory guidelines (e.g. ISO-10993- Biological evaluation of medical devices) which indicated its sufficiency for the VSS's safety evaluation. This approach is aligned with preclinical GLP study design standards, which often rely on small but intensively characterized animal groups to identify major safety signals before clinical translation.
To clarify this point, we have updated the Materials and Methods and Discussion sections accordingly and acknowledged this limitation explicitly.
PRECISE® PRO RX stent, used as reference, is designed for carotid arteries—not cerebral venous sinuses—thus making it an inappropriate comparator. This weakens validity of performance and safety comparisons.
Reply: We appreciate the reviewer’s comment and fully agree that the PRECISE® PRO RX stent was originally designed for use in the carotid arteries. However, this choice was intentional, as the PRECISE stent is among the most commonly used devices in current off-label clinical practice for venous sinus stenting. While it is not optimized for cerebral venous anatomy, it represents the closest selection to what is clinically relevant.
The aim of including the PRECISE stent was not to suggest it is a perfect comparator from a design standpoint, but rather to provide a relevant real-world reference to contextualize the safety of the VIVA Stent System. We have now clarified this rationale in the manuscript.
Use of subjective Likert scale for performance parameters is insufficient in a preclinical device evaluation. Absence of objective, quantitative metrics (e.g., radial force measurements, flow dynamics, endothelial coverage rate) limits reproducibility and weakens the scientific rigor.
Reply: The use of a subjective Likert scale is considered acceptable in preclinical device evaluations. As noted, our approach aligns with the FDA’s 2023 guidance on General Considerations for Animal Studies Intended to Evaluate Medical Devices, which states: “When reporting study findings pertaining to device performance and handling, there should be enough detail to understand the outcomes and potential device safety issues. FDA recommends that a rating scale be created that allows semi-objective scoring of device performance and handling parameters, such as flexibility, pushability, visibility, torquability, kinking, bending, leaking, ease of use, and surgical placement. Rating criteria should encompass steps between the preparation of the device through device placement or use as well as withdrawal and redeployment, if appropriate. We recommend using a semi-quantitative method of assessment for each parameter (e.g., for a scale of 1–4, 1 = poor and 4 = excellent).” It should be noted that Likert-scale based assessments are frequently used in performance evaluations of medical device, and are accepted by regulatory authorities (including frequent indications in published Food and Drug Administration [FDA] guidelines). These types of evaluations offer a quantifiable way of reflecting expert experience in device handling and operation.
Radial force testing was conducted as part of the mechanical evaluation required for stents; however, these data are not reported in the current manuscript. We fully acknowledge the added value of incorporating more objective parameters - such as flow dynamics and endothelial coverage - and have included a statement in the Discussion section to highlight this as a limitation of the present study.
Conclusion claims “excellent safety, performance, and thromboresistance” of VIVA Stent System, suggesting strong support for future clinical use. However, data presented are too limited to support such claims:
Reply: The study design adheres to FDA guidance for preclinical evaluation of medical devices and, together with the results of the required biocompatibility and mechanical testing, provides supportive data for potential future clinical application. However, we acknowledge that the original wording in the conclusion may have overstated the findings. We have therefore revised the conclusion to better reflect the scope and limitations of the current study.
The term "excellent" is not substantiated by quantitative metrics. Conclusion extrapolates efficacy and safety from a small preclinical animal study to human application without appropriate caveats or discussion of limitations. Long-term biological compatibility, a crucial aspect of venous implants, is inadequately discussed given the limited observation time (180 days maximum).
This overreach in interpretation reflects a disconnect between results and conclusion, which may mislead readers about clinical readiness of device.
Reply: We agree that the use of the term “excellent” may overstate the findings, and we have revised the language throughout the manuscript - including in the abstract and conclusions - to adopt a more cautious and evidence-based tone.
We have also added text to the Discussion acknowledging that this study’s 180-day follow-up period, while aligned with regulatory standards for preclinical device evaluation, may not fully capture late restenosis. This aspect warrant dedicated follow-up in future clinical studies.
In this manuscript, there is no indication that authors have engaged with prior work on similar venous stents or cited key clinical trials in this space. High-quality device studies rely on detailed visual documentation, especially for angiographic and histopathologic validation.
Reply: We respectfully note that the manuscript does include references to prior work in this field, particularly studies involving the off-label use of arterial stents for cerebral venous sinus conditions. However, we also emphasize that there is a paucity of high-quality, prospective clinical trials evaluating stents specifically designed for cerebral venous sinus applications. This lack of dedicated devices and robust clinical evidence was in fact a key motivation for developing the VIVA Stent System.
The VIVA Stent System represents a purposeful effort to address this unmet need by creating a device tailored for the anatomical and biomechanical characteristics of the dural venous sinuses. Our study provides the first GLP-compliant in vivo safety and performance data for such a dedicated system.
Regarding visual documentation, we confirm that angiographic and histological assessments were performed in all animals, and representative images are available. These can be provided as supplementary material if requested.
Manuscript lacks novelty, uses suboptimal model and comparator, and overstates the significance of its findings. Methodological weaknesses, combined with exaggerated conclusion not supported by rigorous data. Authors should reconsider their experimental design, articulate clinical relevance more clearly, and adopt more cautious and evidence-driven approach to interpreting results.
Reply: We respectfully disagree with the assertion that the manuscript lacks novelty. To our knowledge, this is the first GLP-compliant in vivo study of a stent specifically developed for use in the cerebral venous system. The rationale for this study stems precisely from the limited availability of dedicated devices and the lack of high-quality clinical evidence in this field, which we have explicitly acknowledged and cited.
While we recognize that the animal model and comparator stent are not perfect analogues for the human intracranial venous system, they were carefully selected based on practical, ethical, and translational considerations. The internal mammary vein in swine allowed for reproducible implantation, controlled safety evaluation, and long-term follow-up, and the PRECISE® stent was chosen as a real-world comparator due to its common off-label use in clinical practice.
We have taken steps throughout the manuscript to temper our conclusions, and emphasize the need for further clinical studies. We believe the data presented provide important first-in-class safety and performance benchmarks that justify continued development of the VIVA Stent System.
We remain committed to evidence-driven research and welcome further dialogue to refine the interpretation of our findings.
Comments on the Quality of English Language
The English could be improved to more clearly express the research
Reply: The manuscript has been carefully reviewed and edited for clarity and language.
Reviewer 3 Report
Comments and Suggestions for Authors
Congratulations for your interesting work regarding this new vein stent. The methodology and the results are well-written and presented. Some suggestions for improvement:
- I would describe the details regarding the technical characteristics of the stent in the methodology and I would avoid such an extended introduction.
- The death of two pigs before the scheduled time due to complications of the vein access rises an issue regarding the device and the injury it can cause to the vascular access. Have you investigated any relevant issues regarding the device? Also, you mention that you had some issues regarding the deployment of the stent for the first three stents. As a consequence, you should consider investigating any technical issues regarding the device, as these issues decrease the technical success of the device. These complications leading to earlier sacrifice of the subjects seem to be serious. I would underline this fact and as a consequence, I would use the word "safe"for the device in a more cautious way.
- In accordance to the previous comment, I would compare the technical succcess of the stent/procedure (primary and secondary) in a seperate paragraph. Please provide a paragraph with the primary and secondary technical success of the device/procedure.
- In the discussion I would compare the stent with data from the literature regarding other vein stents. Generally, please add more studies from the existing literature regarding and comparing to your findings.
- Please provide a paragraph in the discussion regarding the limitations of your study (small number of subjects, animal subjects, technical issues to be studied in the future, etc.)
Author Response
Congratulations for your interesting work regarding this new vein stent. The methodology and the results are well-written and presented. Some suggestions for improvement:
I would describe the details regarding the technical characteristics of the stent in the methodology and I would avoid such an extended introduction.
Reply: We have streamlined the Introduction by removing the detailed technical description of the VIVA Stent System, focusing instead on the clinical rationale. The technical characteristics have been relocated and expanded within the “Test and Reference Items” subsection of the Materials and Methods section (Section 2.2), where they are more appropriate and relevant to the study’s experimental design.
The death of two pigs before the scheduled time due to complications of the vein access rises an issue regarding the device and the injury it can cause to the vascular access. Have you investigated any relevant issues regarding the device? Also, you mention that you had some issues regarding the deployment of the stent for the first three stents. As a consequence, you should consider investigating any technical issues regarding the device, as these issues decrease the technical success of the device. These complications leading to earlier sacrifice of the subjects seem to be serious. I would underline this fact and as a consequence, I would use the word "safe"for the device in a more cautious way.
Reply: Thank you for your comment. There were two occurrences of mortality prior to the first scheduled termination time point (30 days), one animal was found dead and the other was euthanized for humane reasons. Macroscopic findings in the animal that was found dead included pale complexion and diffuse pallor of the liver and spleen, alongside swelling and diffuse intramuscular hematomas in the right hind leg. The designated pathologist determined that the cause of death was procedure-related bleeding due to trauma caused by percutaneous sheath insertion.
The second animal that was euthanized for animal welfare reasons exhibited inability to bear weight on its hindlegs following the procedure. At necropsy, swelling of the right inguinal area accompanied with hematoma within the muscle was noted, which was attributed as well to procedure-related muscular trauma due to the percutaneous sheath insertion, as determined by the designated pathologist.
The femoral vein was never in direct contact with either the delivery system or the stent. The delivery system (loaded with the stent) was introduced through a guiding/judkins right catheter (off-the-shelf products) inserted through the femoral venous sheath and advanced to the internal mammry vein.
Based on the reviewer's comment, we have emphasized the lack of correlation between the observed procedure-related findings in the two cases of early termination, and the Test Item's operation.
As mentioned in our manuscript (lines 270-71), there were some difficulties in detachment of the stent from the delivery system however these difficulties were never associated with any clinical signs or adverse effects, and a geometric change of the grasper component of the delivery system solved this issue.
In accordance to the previous comment, I would compare the technical succcess of the stent/procedure (primary and secondary) in a seperate paragraph. Please provide a paragraph with the primary and secondary technical success of the device/procedure.
Reply: Thank you for this important suggestion. We’ve added a separate paragraph to the Discussion describing the primary and secondary technical success of the device/procedure as follows.
Primary technical success was indicated by the successful deployment of the test stents in the internal mammary veins with full venous wall apposition, and no damage, dissection, perforation or vasospasm to the veins as revealed by intra-procedural fluoroscopy-based evaluations. Pre-terminal computerized tomography-based angiographies of the test stents indicated complete vessel patency as well as complete stent integrity with no fracture, kinks or compression and no stent migration in all study animals. Examination of the test item's delivery system for thrombogenicity, indicated no presence of thrombus on the delivery system's external blood contacting surfaces, thus rendering the VSS as non-thrombogenic under the conditions herein described.
Secondary technical success of the VSS was indicated by the usability and functionality score as determined by three designated interventionalists. Aside from 2 cases graded at 2 (fair & acceptable) in the 180-day cohort, regarding the full detachment of the test stent from the delivery system, all evaluated cases in both cohorts were graded at 4 (very good & acceptable) for all categories. In the course of the initial stenting procedures of the animals assigned to the 180-day cohort, there were sporadic cases in which the test stents did not fully detach from the delivery system. No vascular damage occurred. A geometric modification was made to the grasper component of the delivery system, and these difficulties did not recur in the subsequent stenting procedures performed in animals assigned to the 30-day cohort.
In the discussion I would compare the stent with data from the literature regarding other vein stents. Generally, please add more studies from the existing literature regarding and comparing to your findings.
Reply; A short paragraph was added to the discussion that mentions 2 additional investigational venous stents for cerebral venous stenting (i.e., River stent and BosStent). Information on pre-clinical safety or efficacy studies using these stents is not available in the literature, thus, we can not compare our results to data from the literature regarding the other vein stents.
Please provide a paragraph in the discussion regarding the limitations of your study (small number of subjects, animal subjects, technical issues to be studied in the future, etc.)
Reply: We have added a dedicated paragraph at the end of the Discussion section that outlines the key limitations of the study, including sample size, use of an animal model, and technical considerations for future investigation.
Round 2
Reviewer 2 Report
Comments and Suggestions for Authors
All my suggestions and comments have been addressed.